# Utilization of diabetes management health care services and its association with glycemic control among patients participating in a peer educator-based program in Cambodia

**Mayuree Rao**[1,2]*, **Maurits van Pelt**[3], **James LoGerfo**[2,4], **Lesley E. Steinman**[5], **Hen Heang**[2], **Annette L. Fitzpatrick**[4,6,7]

1 General Medicine Services, VA Puget Sound Health Care System, Seattle, Washington, United States of America, 2 Department of Medicine, University of Washington, Seattle, Washington, United States of America, 3 MoPoTsyo Patient Information Centre, Phnom Penh, Cambodia, 4 Department of Global Health, University of Washington, Seattle, Washington, United States of America, 5 Department of Health Services, University of Washington, Seattle, Washington, United States of America, 6 Department of Family Medicine, University of Washington, Seattle, Washington, United States of America, 7 Department of Epidemiology, University of Washington, Seattle, Washington, United States of America

* mayuree@uw.edu

**Data Availability Statement:** All relevant data are within the manuscript and its supporting information files.

## Abstract

### Background

Substantial evidence supports the effectiveness of peer educator programs for diabetes management in low- and middle-income countries. However, little is known about peer educators' impact relative to other treatment components such as medication and physician consultation. In Cambodia, the non-governmental organization MoPoTsyo organizes four services for people with diabetes: self-management training through peer educator visits, lab tests, physician consultations, and low-cost medicines. Our aims were to 1) quantify MoPoTsyo participant utilization of each program service and 2) define the relationship between each program service and glycemic control.

### Methods

We conducted a retrospective cohort study among 4,210 MoPoTsyo participants, using data collected by MoPoTsyo from 2006–2016. Independent variables assessed were medication adherence, number of peer educator visits, number of physician consultations, and number of lab tests. A multiple logistic regression model was used to evaluate the association between these disease management services and glycemic control—fasting plasma glucose ≤130 mg/dl or post-prandial glucose ≤180 mg/dl—based on most recent glucose level. The model was adjusted for baseline demographic and disease characteristics.

### Findings

Participants with 12 or more peer educator visits per year had a 35% higher odds of glycemic control relative to participants with 4 or fewer visits (odds ratio 1.35, 95% CI: 1.08–1.69;

**Funding:** The authors received no specific funding for this work.

**Competing interests:** Maurits van Pelt and Hen Heang are employed by MoPoTsyo Patient Information Centre, a nonprofit organization. This does not alter our adherence to PLOS ONE policies on sharing data or information.

p = 0.009), after adjustment for utilization of other treatment components (medication adherence, number of physician visits, number of lab tests), follow-up time, and demographic and disease characteristics. Better adherence to medications and a greater number of lab tests per year were also associated with a higher odds of glycemic control after adjustment. Number of physician consultations was not associated with glycemic control after adjustment.

## Conclusions

This study demonstrates a positive association between peer educator utilization and glycemic control incremental to other elements of diabetes management. These results suggest that peer educators may be a valuable addition to comprehensive diabetes management programs in low- and middle-income countries even when other health care services are accessible. The associations identified in this research warrant further prospective studies to explore the causal impact of peer educators on glycemic control relative to other disease management components.

## Introduction

The prevalence of diabetes has increased faster in low- and middle-income countries than in high-income countries over the past four decades [1]. Accordingly, out of an estimated 425 million people with diabetes worldwide, 79% are living in low- and middle-income countries [2]. The disproportionate burden of disease in more resource-limited regions of the world presents multiple challenges to effective diagnosis and management.

Peer educator programs have been used in both high- and low-income countries to improve chronic disease management by providing educational support and linkages to care, particularly in resource-poor settings. A significant body of evidence supports peer educator effectiveness in improving diabetes outcomes [3–6]. However, little is known about the role of peer educators within the broader health care system, perhaps because they are often used where traditional health care services are deficient or difficult to access [5, 7, 8]. The impact of peer educators on diabetes control compared to other important elements of care like physician consultation and medication adherence is largely unknown. Further investigation of the relative benefits of all chronic disease care components, including peer educators, is critical for health system planning in low- and middle-income countries facing a growing burden of noncommunicable disease.

MoPoTsyo Patient Information Center is a Cambodian NGO established in 2004 that has trained peer educators to find and engage members of their communities with diabetes, implement educational sessions on diabetes, and visit these community members monthly to reinforce training and monitor glucose. In addition to its peer educator network, MoPoTsyo also facilitates the following services for its member patients: 1) physician consultations, 2) routine laboratory tests, and 3) low-cost medications through contracted local pharmacies (Revolving Drug Fund). By 2019, MoPoTsyo had trained 255 peer educators and served over 40,000 patients in 8 out of 24 provinces.

In Takeo province, fasting blood glucose and blood pressure were significantly lowered in the first 12 months of participation in MoPoTsyo [9]. However, MoPoTsyo participants do not uniformly use all four program services, and the relative association of each service with glycemic control has never been assessed. Our aims were to 1) quantify MoPoTsyo participant

utilization of each program component and 2) define the relationship between each program component and glycemic control. This analysis may not only reveal opportunities for improvement in MoPoTsyo's population but also advance a more nuanced understanding of the benefit of peer educator programs in resource-poor settings. We achieved these aims as described below.

## Materials and methods

### Program description

MoPoTsyo 's initial mandate was to empower adults with diabetes to better manage their condition by providing disease education and self-management training through community-based peer educators. When MoPoTsyo began, most health centers in Cambodia did not have physicians, and medications for diabetes were not reliably available through the public sector for more than one week at a time. MoPoTsyo evolved to include access to local outpatient physician consultations, laboratory testing, and low-cost medications through a Revolving Drug Fund. Participants pay a small fee for each service used, which sustains the organization and allows peer educators to be compensated for their work. Participants pay a one-time fee of five USD for enrollment, followed by 5.23 USD per month on average for their services [9]. MoPoTsyo's monthly fee represents about five percent of the average monthly household income per capita in Cambodia, which was 102.38 USD in 2016 [10]. Each of MoPoTsyo's services is explained below, but a full description of the program has been published elsewhere [9, 11, 12].

**Peer educators.**   MoPoTsyo selects peer educators among community members with diabetes based on literacy, motivation, and social aptitude. Each peer educator candidate undergoes a six-week training course developed by physicians, pharmacists, and experienced peer educators. This course aims to teach peer educators about the biology of diabetes as well as key components of monitoring and treatment. At the end of the training course, candidates must pass an exam in order to become qualified MoPoTsyo peer educators. Peer educators return to their communities and perform house-to-house diabetes screening. Community members with diabetes are offered enrollment in MoPoTsyo for a one-time fee of five USD which includes educational and self-management materials. Once enrolled, participants with diabetes may attend group sessions—typically monthly—hosted by peer educators in their homes for disease monitoring (point-of-care glucose, blood pressure, and weight), self-management education, and support. Ongoing support and supervision of peer educators is provided by other peer educators in their district and MoPoTsyo staff.

**Physician consultations.**   In order to address the lack of physician capacity, MoPoTsyo recruits physicians (either locally or traveling from Phnom Penh) to provide diabetes consultations, usually once or twice a month, at district hospitals. Peer educators inform MoPoTsyo participants of the scheduled dates that a physician will be providing consultations in their area and collect payment of three USD from those who wish to attend beforehand. On the day of consultation, peer educators also complete patient registration tasks, including vital signs and glucose measurement.

**Revolving drug fund.**   MoPoTsyo has established a Revolving Drug Fund to provide low-cost diabetes and hypertension medications to its members. MoPoTsyo purchases 17 medications in bulk on the international market and sells them to local private and public pharmacies. MoPoTsyo then requires these pharmacies to sell these medications to MoPoTsyo members at a fixed published price per tablet established by MoPoTsyo. The prices of these medications are set such that private pharmacies make a profit based on sale volume. For contracted public

pharmacies, however, an annual monetary reward is calculated based on medication adherence and satisfaction among their MoPoTsyo customers.

**Laboratory tests.** MoPoTsyo provides access to blood and urine tests relevant to diabetes and hypertension (e.g., basic metabolic panel, hemoglobin A1c, lipid panel, and urine protein). Sample collection is organized at local health centers on a fixed schedule, and specimens are transported to MoPoTsyo's central laboratory. Similar to physician consultation days, peer educators are heavily involved; they complete patient registration tasks and provide support and education. Peer educators also explain the lab test results when they are returned in print to members.

## Study design and population

This retrospective cohort study aims to quantify the utilization and association with glycemic control of four MoPoTsyo services: 1) peer educator visits, 2) laboratory tests, 3) physician consultations, and 4) medication adherence. We identified all adults with diabetes enrolled in MoPoTsyo between January 1, 2006 and December 31, 2016. From this cohort, the study population was selected based on pre-specified quality standards required to accurately calculate the outcomes and covariates of interest, described in more detail in the data analysis section below. This study was approved by the University of Washington Human Subjects Division and the National Ethics Committee for Health Research in Cambodia. Both institutional review boards provided waiver of participant consent.

## Data collection

MoPoTsyo maintains an electronic database of members with data collected from encounters with peer educators, physicians, pharmacies, and laboratory. Data are typically recorded on paper forms at the time of each encounter; these paper forms are sent to MoPoTsyo at periodic intervals for entry into a central electronic database. In addition, some contracted pharmacies use a computer-based automated system using barcode readers to record medications dispensed for each member. This data synchronizes with MoPoTsyo's central electronic database.

At initial enrollment, peer educators collect demographic and biologic measures. Peer educators also collect point-of-care capillary glucose using glucometers, blood pressure, and weight at each routine visit with members. Data collected at physician consultations include point-of-care glucose, blood pressure, and medications prescribed. Pharmacy data include invoices of medications purchased (names and prices) by each MoPoTsyo member by date. MoPoTsyo also keeps an annual record of the total quantity and monetary value of medications it has supplied to each pharmacy. Finally, laboratory test results for MoPoTsyo members are also recorded, which include basic metabolic panel, transaminases, hemoglobin A1c, lipid panel, and urine protein. Glucose measurements collected are specified as fasting or postprandial.

## Data analysis

**Baseline characteristics.** We used descriptive statistics including proportions and means with standard deviation (SD) to evaluate the following demographic and biologic characteristics at initial enrollment assessment with MoPoTsyo: age, gender, commune (urban vs rural), comorbid hypertension, symptoms, self-reported medication intake (yes vs no), self-reported years since diabetes diagnosis, and point-of-care glucose. These variables were included in our analysis a priori as potential confounders.

**Medication adherence, peer educator visits, physician consultations, and laboratory tests.** Independent variables in our analysis include medication adherence, number of peer educator visits, number of physician consultations, and number of laboratory tests.

Medication adherence was calculated using Proportion of Days Covered (PDC), a high-quality measure of adherence supported by the Pharmacy Quality Alliance and utilized in the research literature [13, 14]. PDC is the number of days in the follow-up period of interest that are covered by medication divided by the total number of days in the follow-up period, expressed as a percentage. Since medication prices have remained fixed by MoPoTsyo throughout the follow-up period, MoPoTsyo's database calculates and records the expected daily cost of medications prescribed at each physician encounter. This value was compared to the amount spent at each pharmacy encounter in order to calculate the number of days' worth of medication available to each participant, required for determining the numerator of PDC. The denominator of PDC is the total follow-up period for each member. Additional description of PDC calculation is depicted in supplemental Fig 1 (S1 Fig).

We made two assumptions regarding calculation of medication adherence using this method: 1) MoPoTsyo members obtain medications exclusively from pharmacies contracted by MoPoTsyo, and 2) contracted pharmacies are accurately and reliably recording and sending invoices of all medications purchased by MoPoTsyo members for data entry. The first assumption was deemed reasonable given that medications at MoPoTsyo-contracted pharmacies are cheaper than market prices, but there is still likely underestimation of medication purchases since members may obtain medication from many different sources [15]. One source may be a pharmacy not contracted by MoPoTsyo. Additionally, although a prescription from a physician is legally required to dispense these medications in Cambodia, it is possible that some pharmacies or unlicensed sellers inappropriately dispense medications without a prescription; the scale of this phenomenon is unknown [16]. The second assumption was addressed by establishing a data quality threshold for selection of the study population: the annual value of invoices that a contracted pharmacy returned to MoPoTsyo must be $\geq$ 80% of the annual value of medications supplied by MoPoTsyo to that pharmacy, for $\geq$ 80% of the years in which the pharmacy has been participating in MoPoTsyo's Revolving Drug Fund. A priori, MoPoTsyo members with one or more visits to a pharmacy failing to meet this data quality threshold during the follow-up period were not included in the population selected for our analysis.

Number of peer educator visits, physician consultations, and laboratory tests were calculated annually for each MoPoTsyo member.

**Glycemic control.** The dependent variable in our analysis is glycemic control, defined as fasting plasma glucose $\leq$ 130 mg/dl or post-prandial glucose $\leq$ 180 mg/dl, using the most recent glucose level available for each member during the follow-up period. These glycemic targets were chosen based on American Diabetes Association guidelines [17]. The most recent glucose level was point-of-care measurement at a peer educator or physician visit.

Of note, per American Diabetes Association guidelines, post-prandial glucose must be collected one to two hours after a meal in order to capture the peak level. Peer educators are instructed to avoid post-prandial glucose collection within one hour of a meal, but it is possible that some were collected more than two hours after a meal. As a result, some post-prandial glucose levels were likely collected after their peak, thus overestimating glycemic control. However, peer educator-collected glucose levels are the only way to evaluate ongoing glycemic control for many in this resource-poor population, and peer educator visits do not always align with optimal test times. Thus, despite their limitations, post-prandial glucose levels have been included in our analysis to optimize real-world generalizability.

HbA1c levels were not used as the metric for glycemic control in this analysis since they were available for a minority of MoPoTsyo participants. For only 291 participants (6.9%)

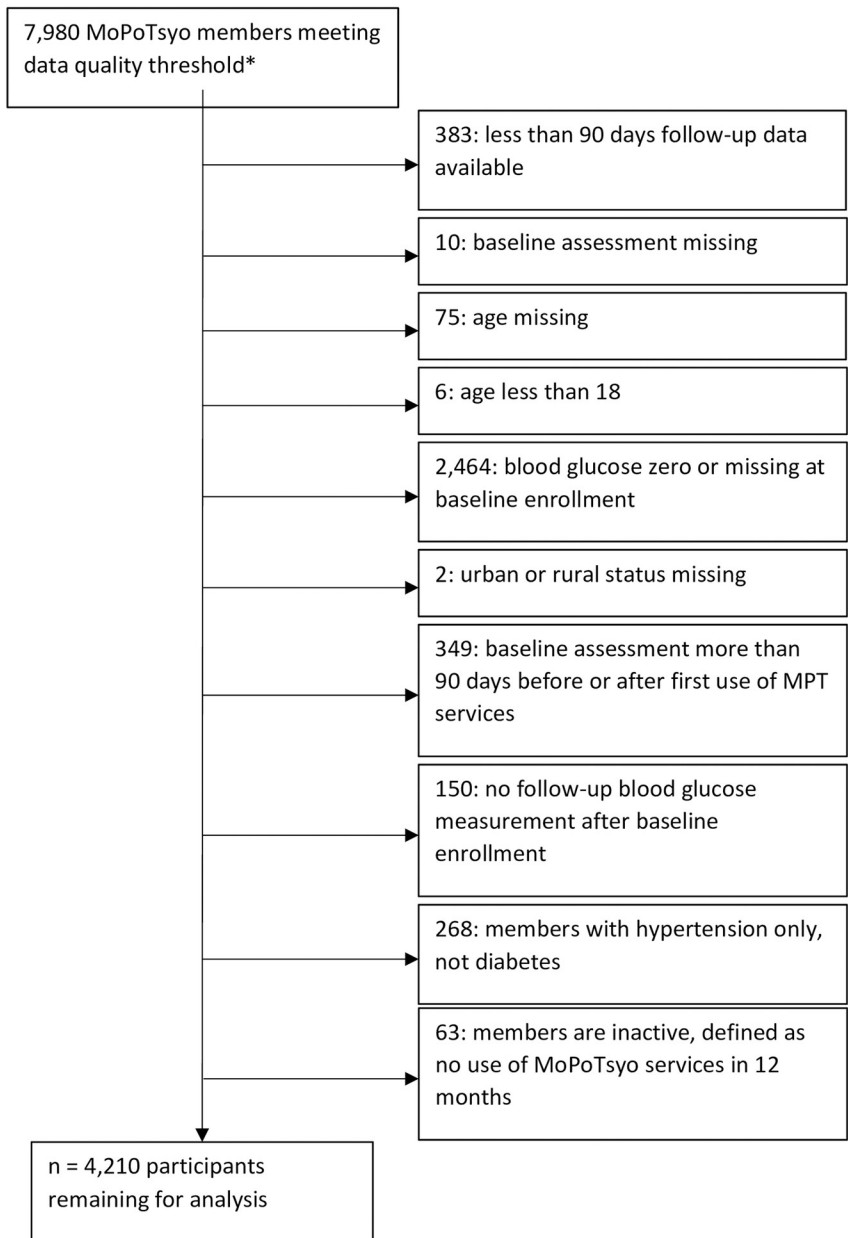

**Fig 1. Study population.**

was an HbA1c level the most recent indicator of glycemic control during the follow-up period.

**Exclusion criteria.** Members were excluded from the analysis if 1) baseline enrollment assessment was not done, 2) one or more baseline characteristics were missing or out of

reasonable range (e.g., glucose level of zero assumed to be measurement or data entry error), 3) participant had hypertension only and not diabetes, 4) less than 90 days of follow-up data was available since medication adherence could not be accurately calculated in these instances [18], 5) baseline enrollment assessment was completed more than 90 days before or after first use of MoPoTsyo services since these data do not reflect characteristics before program participation, or 6) participant had not used any MoPoTsyo services for 12 months, which is MoPoTsyo's definition of an inactive member.

**Statistical methods.** In order to evaluate whether or not the study cohort selected based on the pharmacy data quality threshold was systematically different from the rest of MoPoTsyo's members, we compared these populations' baseline demographic and disease characteristics at enrollment. Differences were assessed using nonparametric and parametric tests as appropriate (eg, t test or Wilcoxon rank-sum test for continuous variables and χ2 or Fisher's exact test for categorical variables) with two-tailed α = 0.05.

We used a multiple logistic regression model to assess the degree to which medication adherence, number of peer educator visits, number of physician consultations, and number of laboratory tests were associated with glycemic control. Models were adjusted for pre-specified baseline demographic and disease characteristics described above as well as follow-up period. All analyses were conducted in Stata version 15 (StataCorp, College Station, Texas) using command logistic with two-tailed α = 0.05.

## Results

### Inclusion/exclusion

A total of 7,980 members of MoPoTsyo between Jan 1, 2006 and December 31, 2016 were selected as the study population based on the pharmacy data quality criterion. From this initial cohort, 2,707 participants were excluded for having one or more baseline covariates missing or out of range. 383 participants with less than 90 days of follow-up data were excluded. 349 participants whose first use of a MoPoTsyo service was more than 90 days before or after baseline assessment were excluded. 268 participants with hypertension but not diabetes were excluded. 63 inactive members (no use of any MoPoTsyo services for 12 months) were excluded. 4,210 participants were included in the final analysis (Fig 1).

### Baseline characteristics

Baseline demographic and biologic characteristics are presented in Table 1 for the study population. At the time of enrollment in MoPoTsyo, mean age of the study population was 55 years (SD 11). Sixty-five percent were female and 54% lived in rural households. Fifty-three percent had co-morbid hypertension, 90% had at least one symptom of diabetes or hypertension, 42% reported already taking a diabetes or hypertension medication, and the mean time since diabetes diagnosis was 2.6 years (SD 4.1). Most participants had poorly controlled glucose at enrollment (53% with fasting plasma glucose over 200). Mean follow-up period was 3.4 years (SD 1.6).

MoPoTsyo participants not included in the initially selected study population (n = 10,558) due to failure to meet the pharmacy data quality threshold were more likely to be female (67% compared to 65%, p = 0.032), live in a rural commune (82% compared to 54%; p<0.0005), and have good glycemic control at enrollment (17% compared to 9%; p<0.0005). They were less likely to have symptoms (86% compared to 90%; p<0.0005) or take medication at enrollment (34% relative to 42%; p<0.0005). They had a shorter mean time since diabetes diagnosis (2.1 relative to 2.6 years; p<0.00005) and follow-up period (3.3 relative to 3.4 years; p<0.00005).

**Table 1. Baseline characteristics of study population.**

| Baseline characteristics | Study population, n (%) |
|---|---|
| Age, years[*] | 55 (11) |
| Female | 2,752 (65) |
| Rural | 2,288 (54) |
| Good glycemic control[†] | 373 (9) |
| Fasting plasma glucose, mg/dL[‡] | |
| $\leq 150$ | 831 (20) |
| 151 to 200 | 1,166 (28) |
| > 200 | 2,131 (52) |
| Comorbid hypertension | 2,233 (53) |
| At least one symptom of diabetes or hypertension[¥β] | 3,794 (90) |
| Takes diabetes or hypertension medication[β] | 1,784 (42) |
| Years since diagnosis with diabetes[*β€] | 2.6 (4.1) |
| Follow up period, years[*] | 3.4 (1.6) |
| **Total** | 4,210 (100) |

[*]mean (standard deviation).

[†]Defined as $\leq 130$ fasting or $\leq 180$ post-prandial, from point-of-care glucose measurement.

[¥]One of the following as self-reported: tingling, numbness, burning feet, ulcer, chest pain, neck pain, blurry vision, headache, dizziness.

[β] Self-reported.

[‡]Missing for 82 subjects.

[€]Missing or out of range (e.g., greater than age) for 244 subjects, not included in mean.

The mean age and incidence of co-morbid hypertension was about the same as the study population.

## Program component utilization

Table 2 presents descriptive statistics of independent variables medication adherence, number of peer educator visits, number of physician consultations, and number of laboratory tests during the follow-up period. Medication adherence (defined as PDC) was 0–19% for 46% of the study population, 20–39% for 14% of the population, 40–59% for 13% of the population, 60–79% for 14% of the population, and 80–100% for 13% of the population. Mean medication adherence was 34.1% (SD 32.2%). Sixty-three percent had 4 or fewer peer educator visits per year, 50% had less than one physician consultation per year, and 46% had between 0 and 1 laboratory test per year.

## Glycemic control

The number of study participants with good glycemic control based on the most recently collected glucose during the follow-up period was 1,400 (33.3%). Table 3 presents multiple logistic regression models of program components associated with good glycemic control. In the first model, each program component variable was adjusted for baseline enrollment demographic and disease characteristics as well as follow-up period. All program component variables except for number of physician consultations were associated with statistically significant higher odds of glycemic control. Members with medication adherence of 60–79% and 80–100% had statistically significant higher odds of glycemic control relative to those with medication adherence 0–19% (OR 1.30; 95% CI: 1.05, 1.60; p = 0.014 for adherence 60–79% and OR

**Table 2. Utilization of MoPoTsyo program components during follow-up period in study population.**

| Program component | Study population, n (%) |
|---|---|
| Number of peer educator visits per year | |
| ≤ 4 | 2,640 (63) |
| 5 to 11 | 788 (19) |
| ≥ 12 | 782 (19) |
| Number of clinician visits per year | |
| < 1 | 2,117 (50) |
| 1 to 3 | 1,571 (37) |
| ≥ 4 | 522 (12) |
| Number of lab tests per year | |
| 0 | 1,226 (29) |
| > 0 and < 1 per year | 1,954 (46) |
| ≥ 1 per year | 1,030 (25) |
| Medication adherence (proportion of days covered, %) | |
| 0–19 | 1,932 (46) |
| 20–39 | 607 (14) |
| 40–59 | 540 (13) |
| 60–79 | 586 (14) |
| 80–100 | 545 (13) |
| **Total** | **4,210 (100)** |

1.45; 95% CI: 1.18, 1.79; p = 0.001 for adherence 80–100%). Although medication adherence of 20–39% and 40–59% were also associated with a higher odds of glycemic control, the odds ratios did not reach statistical significance. Twelve or more peer educator visits per year was associated with 51% higher odds of glycemic control relative to four or less visits per year (OR 1.51; 95% CI: 1.22, 1.85; p < 0.0005). A greater number of laboratory tests per year was associated with higher odds of glycemic control, with OR 1.35 (95% CI: 1.14, 1.60; p = 0.001) for greater than zero but less than one test per year and OR 1.70 (95% CI: 1.38, 2.09; p < 0.0005) for one or more test per year compared to reference category of zero tests per year. Finally, at least one physician consultation per year was associated with higher odds of glycemic control compared to less than one per year, but the odds ratios did not reach statistical significance.

After adding all program component variables to the model—keeping the adjustment for baseline demographic/disease characteristics and follow-up period—the number of peer educator visits, medication adherence, and number of laboratory tests all remained statistically significantly associated with glycemic control. Twelve or more peer educator visits per year was associated with 35% higher odds of glycemic control (OR 1.35; 95% CI: 1.08, 1.69; p = 0.009). Medication adherence 80–100% was associated with 34% higher odds of glycemic control (OR 1.34; 95% CI 1.03, 1.73; p = 0.027). Greater than zero but less than one laboratory test per year was associated with 33% higher odds of glycemic control (OR 1.33; 95% CI: 1.12, 1.59; p = 0.001) and one or more test per year with 56% higher odds of glycemic control (OR 1.56; 95% CI: 1.20, 2.02; p = 0.001). Like the previous model, physician consultations were not associated with glycemic control after adjustment for all baseline covariates and other treatment adherence variables.

## Discussion

Among the MoPoTsyo program components examined, we found that medication adherence, peer educator visits, and laboratory tests were individually significantly associated with

**Table 3. Logistic regression of association between independent variables defined as utilization of MoPoTsyo program components and dependent variable glycemic control (≤ 130 fasting or ≤ 180 post-prandial).**

| Covariate | OR adjusted for baseline covariates (95% CI)[*] | p value[†] | OR adjusted for baseline covariates and all other MoPoTsyo program components (95% CI) | p value |
|---|---|---|---|---|
| Medication adherence | | | | |
| 0–19% (reference) | -- | | -- | |
| 20–39% | 1.09 (0.88, 1.34) | 0.424 | 1.04 (0.84, 1.30) | 0.705 |
| 40–59% | 1.17 (0.95, 1.45) | 0.147 | 1.07 (0.84, 1.36) | 0.597 |
| 60–79% | **1.30 (1.05, 1.60)** | **0.014** | 1.16 (0.90, 1.49) | 0.243 |
| 80–100% | **1.45 (1.18, 1.79)** | **0.001** | **1.34 (1.03, 1.73)** | **0.027** |
| Number of peer educator visits per year | | | | |
| ≤ 4 (reference) | -- | | -- | |
| 5 to 11 | 1.13 (0.94, 1.35) | 0.199 | 1.09 (0.90, 1.31) | 0.366 |
| ≥ 12 | **1.51 (1.22, 1.85)** | **< 0.0005** | **1.35 (1.08, 1.69)** | **0.009** |
| Number of clinician visits per year | | | | |
| < 1 (reference) | -- | | -- | |
| 1 to 3 | 1.11 (0.96, 1.30) | 0.160 | 0.87 (0.72, 1.05) | 0.143 |
| ≥ 4 | 1.25 (0.99, 1.57) | 0.059 | 0.76 (0.56, 1.04) | 0.085 |
| Number of lab tests per year | | | | |
| 0 (reference) | -- | | -- | |
| > 0 and < 1 per year | **1.35 (1.14, 1.60)** | **0.001** | **1.33 (1.12, 1.59)** | **0.001** |
| ≥ 1 per year | **1.70 (1.38, 2.09)** | **< 0.0005** | **1.56 (1.20, 2.02)** | **0.001** |
| Age at baseline | | | **1.01 (1.00, 1.02)** | **0.003** |
| Female gender | | | **0.86 (0.75, 0.99)** | **0.038** |
| Rural household | | | **1.23 (1.03, 1.46)** | **0.020** |
| Good glycemic control at baseline[¥] | | | **2.08 (1.66, 2.61)** | **< 0.0005** |
| Comorbid hypertension | | | 0.93 (0.81, 1.08) | 0.339 |
| At least one symptom of diabetes or hypertension at baseline[£β] | | | 0.91 (0.73, 1.15) | 0.439 |
| Takes medication at baseline[β] | | | **0.84 (0.73, 0.98)** | **0.023** |
| Years since diagnosis with diabetes at baseline[β‡] | | | 1.00 (1.00, 1.00) | 0.750 |
| Total follow-up period, years | | | 0.97 (0.92, 1.02) | 0.180 |

[*]Logistic regression of association between utilization of each MoPoTsyo program component (independent variable) and glycemic control (dependent variable). Each model is adjusted for the baseline covariates in the table, but not for other MoPoTsyo program components. Glycemic control is based on the most recently collected glucose measurement (either point-of-care or laboratory value) for each participant during the follow-up period.

[†]Bolded p-values and confidence intervals are statistically significant, pre-specified as p < 0.05.

[¥]Defined as ≤ 130 fasting or ≤ 180 post-prandial, from point-of-care glucose measurement.

[£]One of the following as self-reported: tingling, numbness, burning feet, ulcer, chest pain, neck pain, blurry vision, headache, dizziness.

[β]Self-reported.

[‡]Missing or out of range (e.g., greater than age) for 250 subjects.

glycemic control after adjustment for baseline enrollment demographic and disease covariates as well as follow-up period. Physician consultations were also associated with glycemic control, but this effect size did not reach statistical significance.

A greater number of peer educator visits was still associated with improved glycemic control even after adjustment for baseline demographic/disease covariates, follow-up period, and utilization of other program components (medication adherence, physician consultations, and laboratory services). Medication adherence remained statistically significantly associated with

glycemic control after full adjustment, as expected for a pillar of diabetes control. Of note, utilization of laboratory tests also remained significantly associated with glycemic control after full adjustment. It is possible that participants who utilize laboratory tests are more likely to have good glycemic control due to another shared characteristic, such as higher socioeconomic status given the additional cost. Another possible explanation is that MoPoTsyo-organized lab services are community events including social support and education from peer educators. Furthermore, printed lab results are returned to members in the Khmer language with additional explanation provided by peer educators, further facilitating appropriate diabetes care. The strength of the peer educator effect in this analysis may be understood in the context of MoPoTsyo's program model in which peer educators are integrally involved in organizing and implementing other program services.

Although physician consultations were not associated with glycemic control after full adjustment for other treatment components, this result is likely due to the strong association between physician consultation and medication adherence in the Cambodian context. Physician consultations organized by MoPoTsyo are typically short visits for the purpose of medication prescription and adjustment. In a prior analysis of medication adherence in MoPoTsyo's population, four or more physician consultations per year was associated with an increase in medication adherence by 47 percentage points, even after adjustment for baseline covariates [19]. In the present analysis, the Pearson correlation coefficient between medication adherence and number of physician consultations per year was 0.58 ($p < 0.00005$). Given the collinearity between medication adherence and physician consultation, the impact of each on glycemic control may be expected to diminish after adjustment for each other. However, given a priori selection and improved overall model fit, both variables were kept in the fully adjusted analysis.

Of note, utilization of all four components of MoPoTsyo's program was low. The majority of participants had less than or equal to four peer educator visits per year, less than one clinician visit per year, less than one laboratory test per year, and medication adherence 0–19%. Our study analyzed utilization across the entire membership period for each participant, with a mean follow-up period of 3.4 years. It is plausible that utilization of program services was higher after initial enrollment and waned later in the membership. This hypothesis merits future investigation in order to better characterize the initial versus longer-term effects of peer educator programs.

The positive association between peer educator visits and glycemic control identified in this study reflects similar findings in studies of community health worker programs for patients with diabetes in low- and middle-income countries including India, Iran, Jamaica, Guatemala, and American Samoa [3–5, 20–26].

The primary strength of this study is its analysis of the association between peer educator visits and glycemic control relative to other standard components of disease management. While other community health worker programs may also include linkages to clinical providers, laboratory tests, and/or medications, few have published data separating the effects of each of these interventions on glycemic control [4]. As a result, the current literature rarely examines the effectiveness of community health workers in the context of the broader health care system [5]. Our study contributes to the evidence by addressing this limitation. Additionally, as far as the authors are aware, this paper presents one of the only analyses of medication adherence in the context of a Revolving Drug Fund created by a non-governmental organization in a resource-poor setting. Finally, we believe these results may be relevant to other resource-poor populations with a growing prevalence of diabetes.

This study had some limitations. First, it is likely that medication adherence was underestimated due to both missing pharmacy invoices and the use of non MoPoTsyo- associated physician consultations and medicines by study participants. We attempted to address the former by imposing a pharmacy data quality threshold for the selection of the study population. Participants not selected for failing to meet this quality threshold overall had significantly different baseline demographic and disease characteristics than participants selected for our analysis. Although we adjusted for these covariates in our analysis, this data quality issue may have increased the strength of the association between peer educator visits/laboratory tests and glycemic control. For the latter limitation, since medicines at MoPoTsyo-contracted pharmacies are cheaper than market prices, we hypothesize that the number of participants purchasing medicines elsewhere is few. However, we are unable to identify participants who purchased medicines from non MoPoTsyo-contracted pharmacies and/or unlicensed sellers and therefore cannot systematically account for this issue. Medication adherence is likely underestimated as a result. Second, it is possible that the association between peer educator visits and glycemic control can be explained by other participant characteristics not accounted for in this analysis; although most clinically relevant confounders were included in the model, socioeconomic status, educational level, and family characteristics were key covariates not available in our data. Third, the use of post-prandial glucose levels (when fasting levels were not available) as a measure of glycemic control is limited by variability in the timing of collection by peer educators. Specifically, peer educators may collect post-prandial glucose more than two hours after a meal, leading to overestimation of glycemic control in this analysis. However, inclusion of post-prandial glucose levels enhances real-world generalizability to other resource-poor settings in which home glucose collection at optimal times is not always possible. Fourth, this study may not be generalizable to a non-Cambodian population or to other peer educator programs, but we believe the results have relevance to chronic disease management in other resource-poor settings.

This study examines the association between utilization of four components of a peer educator-based diabetes management program in Cambodia and glycemic control: 1) peer educator visits, 2) physician consultations, 3) laboratory tests, and 4) medication adherence through a Revolving Drug Fund. We believe the most significant result is that a greater number of peer educator visits was found to be associated with glycemic control, incremental to the utilization of other program components. Prospective studies are necessary to establish a causal relationship, but these findings suggest that peer educator programs may provide significant additional benefit to diabetes control even when other standard chronic disease care elements like medications and physician consultations are available. The integration of peer educators into health systems merits consideration in low- and middle-income countries facing a growing burden of non-communicable disease.

## Supporting information

**S1 Fig. Proportion of days covered.** Calculation of the medication adherence measure used in study, Proportion of Days Covered, using a hypothetical follow-up period and purchase of medications.
(DOCX)

**S1 File. Dataset for study analysis.** Dataset used to calculate baseline characteristics of study population and logistic regression model analyzing association between utilization of MoPoTsyo services and glycemic control.
(DTA)

## Author Contributions

**Conceptualization:** Mayuree Rao, Maurits van Pelt, James LoGerfo, Lesley E. Steinman, Annette L. Fitzpatrick.

**Data curation:** Mayuree Rao, Maurits van Pelt, Hen Heang.

**Formal analysis:** Mayuree Rao.

**Methodology:** Mayuree Rao, Maurits van Pelt, James LoGerfo, Lesley E. Steinman, Annette L. Fitzpatrick.

**Writing – original draft:** Mayuree Rao.

**Writing – review & editing:** Mayuree Rao, Maurits van Pelt, James LoGerfo, Lesley E. Steinman, Annette L. Fitzpatrick.

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
