## [Decision Letter · Decision Letter 0]

24 Mar 2020

PONE-D-20-00770

Treatment adherence among patients with diabetes participating in a peer educator-based disease management program in Cambodia

PLOS ONE

Dear Dr. Rao,

Thank you for submitting your manuscript to PLOS ONE. After careful consideration, we feel that it has merit but does not fully meet PLOS ONE’s publication criteria as it currently stands. Therefore, we invite you to submit a revised version of the manuscript that addresses the points raised during the review process.

We would appreciate receiving your revised manuscript by May 08 2020 11:59PM. To enhance the reproducibility of your results, we recommend that if applicable you deposit your laboratory protocols in protocols.io, where a protocol can be assigned its own identifier (DOI) such that it can be cited independently in the future. For instructions see: http://journals.plos.org/plosone/s/submission-guidelines#loc-laboratory-protocols

We look forward to receiving your revised manuscript.

Kind regards,

Siyan Yi, MD, MHSc, PhD

Academic Editor

PLOS ONE

Journal Requirements:

Reviewers' comments:

Reviewer's Responses to Questions

**Comments to the Author**

1. Is the manuscript technically sound, and do the data support the conclusions?

Reviewer #1: Yes

Reviewer #2: Partly

Reviewer #3: Yes

2. Has the statistical analysis been performed appropriately and rigorously? 

Reviewer #1: Yes

Reviewer #2: Yes

Reviewer #3: Yes

3. Have the authors made all data underlying the findings in their manuscript fully available?

Reviewer #1: Yes

Reviewer #2: Yes

Reviewer #3: Yes

4. Is the manuscript presented in an intelligible fashion and written in standard English?

Reviewer #1: Yes

Reviewer #2: Yes

Reviewer #3: Yes

5. Review Comments to the Author

Reviewer #1: Major comments

#1 Glycemic control: I have a major concern on the measure of glycemic control in this study. First, I have doubted about the accuracy of post-prandial glucose that should be measured 1–2 h after the beginning of the meal as recommended by ADA, generally peak levels in patients with diabetes. The variation in measure this parameter, specifically not within the recommended time range may occur in the practice of peer educators. Second, I wonder why the authors did not consider HbA1C as an indicator for glycaemic control even lab test for this parameter were also provided to MoPoTsyo members.

#2 Medication adherence: I appreciate the authors attempt to validate data from contracted pharmacies by setting a data quality threshold. However, as the authors mentioned, MoPoTsyo members may purchase their medicines from elsewhere other than contracted pharmacies. The likelihood of underestimated adherence may affect the final model and should not be overlooked.

Minor comments

#3 Title: Please consider revise the title, especially the words “treatment adherence”, since it might mislead the readers and does not reflect the context of this study on the four components of MoPoTsyo program.

#4 Abstract: The objective of this study should be clearly defined and consistent with the main texts.

#5 Participant fee: Please clarify if the participant fee for MoPoTsyo program was 5.23 USD per month (Page 6, Line 95), or 5 USD one-time (Page 6, Line 104).

#6 Results: Please present findings of final glycemic control as proportion of participants for each glycemic range.

#7 Discussion: I think it would be more interesting to compare findings from this study with others that explore the effects of peer educator to T2D patients, or to other medical conditions in LMIC.

#8 Discussion: Since adherence to all four components of MoPoTsyo program of nearly half of the participants were generally low, it would be nice if the authors could provide some speculations on this phenomenon.

Reviewer #2: This is primarily aimed to examine the treatment adherence among patients with diabetes participating in a peer educator-based disease management program in Cambodia. Secondarily, it examined the impact of treatment adherence on glycaemic control. By looking at these study objectives, the study design employed (retrospective cohort study) could not provide answer to the secondary objective. This is because the findings generated from a retrospective single cohort study without having a comparative cohort could not draw any conclusion on the causal effect between the Program and the clinical outcome (glycaemic control). Hence, it is strongly suggested that authors should replace the word impact with association throughout the manuscript. For example, the conclusions mentioned that “This study demonstrates a positive impact of peer educators on glycemic control incremental to other elements of diabetes treatment.” The word impact should be replaced by association. “These results strengthen the evidence for peer educators as a valuable component of comprehensive diabetes management programs in low- and middle-income countries.” I do not agree that this is your conclusion since you can not conclude the impact of the Program based on your study design. Please rephrase.

Reviewer #3: It is an important piece of analysis to support the role of peer educators in diabetes disease management. However, I have a number of questions.

1. Regarding glycemic control, why didn't the authors use HbA1c level? I think it is more accurate to indicate good control of blood glucose.

2. Are number of peer educator visits, number of physician consultations, and number of laboratory tests, medication adherence really independent from one another?

3. Related to your first study limitation, is there any possibility that a patient get medicines without having to visit peer educators, consult physicians, or have lab tests?

4. How are the four independent variables related to self-management (that might lead to glycemic control)? I am worried about influence of demographic and disease characteristics on the findings. As a matter of fact, self-management can be attributable to many other variables such as both family and non-family support, educational level, current family responsibility (role in the family), socio-economic status,...etc.

The conclusion addresses the health system, but the peer educator network does not seem to work with health centers (backbone of the primary health care).

I also notice that the discussion section is so contextualized and does not relate the findings to studies in other low-and middle income countries. The explanation of lab services (their statistical significance) is not quite convincing to me (based on my field observation and experience).

I am not quite clear of the "Greater than zero but less than one laboratory test per year". What does it mean?

6. PLOS authors have the option to publish the peer review history of their article (what does this mean?). If published, this will include your full peer review and any attached files.

Reviewer #1: No

Reviewer #2: No

Reviewer #3: No

---

## [Author Response · Author response to Decision Letter 0]

8 May 2020

Reviewer #1: Major comments

#1 Glycemic control: I have a major concern on the measure of glycemic control in this study. First, I have doubted about the accuracy of post-prandial glucose that should be measured 1–2 h after the beginning of the meal as recommended by ADA, generally peak levels in patients with diabetes. The variation in measure this parameter, specifically not within the recommended time range may occur in the practice of peer educators. Second, I wonder why the authors did not consider HbA1C as an indicator for glycaemic control even lab test for this parameter were also provided to MoPoTsyo members.

Response: MoPoTsyo’s peer educators are instructed that glucose collection within 1 hour of a meal does not reflect glycemic control, and participants are educated as such too. Per discussion with MoPoTsyo leadership, it is highly unlikely that peer educators are collecting point-of-care glucoses within one hour of meals. However, it is possible that peer educators are collecting and recording point-of-care glucose levels more than 2 hours after meals. As such, by using the standard ADA threshold of 180 for post-prandial glucose, we are probably overestimating glycemic control. On the other hand, like many resource-poor settings, participants don’t necessarily have their own glucometers and testing supplies to collect their own home glucose levels at the correct time. The glucose levels collected at their peer educator follow-up visits may be their only ongoing way of assessing glycemic control, and the timing of these visits may not always coincide with optimal test times. The inclusion of post-prandial glucoses in our analysis favors generalizability to other resource-poor populations, at the expense of overestimating glycemic control. We have included additional language in the manuscript to describe these issues in the Data Analysis section and Discussion section.

As for HbA1c, we agree that this would be ideal, but too few participants have a recent HbA1c. As we described in the manuscript, for the outcome glycemic control, we used the most recent glucose level available for each member during the follow-up period. For only 291 participants was the most recent indicator of glycemic control an HbA1c level, even when we include HbA1c levels drawn within one month prior to the most recent glucose level. At the time of data collection, the price of the HbA1c lab test was USD 7.50, while the price of the point-of-care glucose test was USD 0.63 if collected by the peer educator. Given that MoPoTsyo serves economically deprived urban and rural areas, this price difference likely explains the relatively few HbA1cs performed in this population. We have included additional language in the manuscript to explain our use of glucose rather than HbA1c as the indicator of glycemic control.

#2 Medication adherence: I appreciate the authors attempt to validate data from contracted pharmacies by setting a data quality threshold. However, as the authors mentioned, MoPoTsyo members may purchase their medicines from elsewhere other than contracted pharmacies. The likelihood of underestimated adherence may affect the final model and should not be overlooked.

Response: We certainly agree with this concern and chose to highlight this issue as our first limitation in the discussion section. In the revised manuscript, we have added additional language in the discussion section to further emphasize this limitation.

Minor comments

#3 Title: Please consider revise the title, especially the words “treatment adherence”, since it might mislead the readers and does not reflect the context of this study on the four components of MoPoTsyo program.

Response: Revised title: Utilization of diabetes management health care services and its association with glycemic control among patients participating in a peer educator-based program in Cambodia

#4 Abstract: The objective of this study should be clearly defined and consistent with the main texts.

Response: We revised the background section of the abstract to include an aims statement and used the same aims statement in the main text, as below:

“Our aims were to 1) quantify MoPoTsyo participant utilization of each program service and 2) define the relationship between each program service and glycemic control.”

#5 Participant fee: Please clarify if the participant fee for MoPoTsyo program was 5.23 USD per month (Page 6, Line 95), or 5 USD one-time (Page 6, Line 104).

Response: Participants pay 5 USD one-time for enrollment, followed by 5.23 USD per month. We have revised the manuscript to clarify the fees.

#6 Results: Please present findings of final glycemic control as proportion of participants for each glycemic range.

Response: We have added the findings of final glycemic control as a proportion of participants to the results section, under the sub-section “Glycemic control,” as below.

“The number of study participants with good glycemic control based on the most recently collected glucose during the follow-up period was 1,400 (33.3%).”

#7 Discussion: I think it would be more interesting to compare findings from this study with others that explore the effects of peer educator to T2D patients, or to other medical conditions in LMIC.

Response: We have added some additional language to the discussion section to more clearly compare/contrast our study with other studies on peer educators for diabetes patients, as below.

“The positive association between peer educator visits and glycemic control identified in this study reflects similar findings in studies of community health worker programs for patients with diabetes in low- and middle-income countries including India, Iran, Jamaica, Guatemala, and American Samoa [3-5, 16-22]. The primary strength of this study is its analysis of the association between peer educator visits and glycemic control relative to other standard components of disease management. While other community health worker programs may also include linkages to clinical providers, laboratory tests, and/or medications, few have published data separating the effects of each of these interventions on glycemic control [4]. As a result, the current literature rarely examines the effectiveness of community health workers in the context of the broader health care system [5]. Our study contributes to the evidence by addressing this limitation.”

#8 Discussion: Since adherence to all four components of MoPoTsyo program of nearly half of the participants were generally low, it would be nice if the authors could provide some speculations on this phenomenon.

Response: We added the following language to the discussion section.

“Of note, utilization of all four components of MoPoTsyo’s program was low. The majority of participants had less than or equal to four peer educator visits per year, less than one clinician visit per year, less than one laboratory test per year, and medication adherence 0-19%. Our study analyzed utilization across the entire membership period for each participant, with a mean follow-up period of 3.4 years. It is plausible that utilization of program services was higher after initial enrollment and waned later in the membership. This phenomenon merits future study in order to better characterize the initial versus longer-term effects of peer educator programs.”

Reviewer #2: This is primarily aimed to examine the treatment adherence among patients with diabetes participating in a peer educator-based disease management program in Cambodia. Secondarily, it examined the impact of treatment adherence on glycaemic control. By looking at these study objectives, the study design employed (retrospective cohort study) could not provide answer to the secondary objective. This is because the findings generated from a retrospective single cohort study without having a comparative cohort could not draw any conclusion on the causal effect between the Program and the clinical outcome (glycaemic control). Hence, it is strongly suggested that authors should replace the word impact with association throughout the manuscript. For example, the conclusions mentioned that “This study demonstrates a positive impact of peer educators on glycemic control incremental to other elements of diabetes treatment.” The word impact should be replaced by association. “These results strengthen the evidence for peer educators as a valuable component of comprehensive diabetes management programs in low- and middle-income countries.” I do not agree that this is your conclusion since you can not conclude the impact of the Program based on your study design. Please rephrase.

Response: We have revised the entire manuscript, from abstract to discussion, to clearly state that we have identified an association, rather than impact. We also revised our aims to improve specificity and clarity.

Reviewer #3: It is an important piece of analysis to support the role of peer educators in diabetes disease management. However, I have a number of questions.

1. Regarding glycemic control, why didn't the authors use HbA1c level? I think it is more accurate to indicate good control of blood glucose.

Response: See response to similar comment above, copied again below.

As for HbA1c, we agree that this would be ideal, but too few participants have a recent HbA1c. As we described in the manuscript, for the outcome glycemic control, we used the most recent glucose level available for each member during the follow-up period. For only 291 participants was the most recent indicator of glycemic control an HbA1c level, even when we included HbA1c levels drawn within one month prior to the most recent glucose level. At the time of data collection, the price of the HbA1c lab test was USD 7.50, while the price of the point-of-care glucose test was USD 0.63 if collected by the peer educator. Given that MoPoTsyo serves economically deprived urban and rural areas, this price difference likely explains the relatively few HbA1cs performed in this population. We have included additional language in the manuscript to explain our use of glucose rather than HbA1c as the indicator of glycemic control.

2. Are number of peer educator visits, number of physician consultations, and number of laboratory tests, medication adherence really independent from one another?

Response: The Pearson correlation coefficients between all combinations of our independent variables are listed below.

Medication adherence and number of physician consultations = 0.5823

Medication adherence and number of peer educator visits = 0.3126

Medication adherence and number of laboratory tests = 0.5529

Number of peer educator visits and number of laboratory tests = 0.5549

Number of peer educator visits and number of physician consultations = 0.4353

Number of laboratory tests and number of physician consultations = 0.6169

We discuss the correlation between medication adherence and number of physician consultations in the manuscript already, in the context of our finding that physician consultations were not associated with glycemic control after adjustment for other program components. As above, there is indeed moderate correlation between all of our independent variables. However, given a priori selection and improved overall model fit, all variables were included in our final adjusted analysis.

3. Related to your first study limitation, is there any possibility that a patient get medicines without having to visit peer educators, consult physicians, or have lab tests?

Response: It is legally required to have a physician prescription to obtain a medication at a pharmacy. Peer educator visits or laboratory tests are not required. We have added the bolded following language to the methods section to explain.

“We made two assumptions regarding calculation of medication adherence using this method: 1) MoPoTsyo members obtain medications exclusively from pharmacies contracted by MoPoTsyo, and 2) contracted pharmacies are accurately and reliably recording and sending invoices of all medications purchased by MoPoTsyo members for data entry. The first assumption was deemed reasonable given that medications at MoPoTsyo-contracted pharmacies are cheaper than market prices, but there is still likely underestimation of medication purchases since members may obtain medication from many different sources [14]. One source may be a pharmacy not contracted by MoPoTsyo. Additionally, although a prescription from a physician is legally required to dispense these medications in Cambodia, it is possible that some pharmacies or unlicensed sellers inappropriately dispense medications without a prescription; the scale of this phenomenon is unknown [15].”

4. How are the four independent variables related to self-management (that might lead to glycemic control)? I am worried about influence of demographic and disease characteristics on the findings. As a matter of fact, self-management can be attributable to many other variables such as both family and non-family support, educational level, current family responsibility (role in the family), socio-economic status,...etc.

Response: We adjusted for demographic and disease characteristics in both logistic regression models reported in the study. The characteristics we adjusted for include age, gender, rural status, comorbid hypertension, glycemic control at baseline enrollment, symptoms of diabetes or hypertension at baseline, time since diagnosis, whether or not participant takes medication at baseline enrollment, and length of follow-up period. We hoped by adjusting for these variables that we would control for many elements that contribute to self-management. However, we were not able to adjust for every variable that may contribute to self-management, as you point out. We mentioned the absence of socio-economic status as an important limitation to this study. We have now added language to also mention the absence of educational level and family characteristics in the model as limitations. 

The conclusion addresses the health system, but the peer educator network does not seem to work with health centers (backbone of the primary health care).

Response: The ministry of health in Cambodia did not provide a clinical guideline for Health Centers to provide diabetes and hypertension care until 2019. MoPoTsyo’s model has been part of the discussion as the Ministry of Health continues to develop its chronic care policies and systems.

I also notice that the discussion section is so contextualized and does not relate the findings to studies in other low-and middle income countries. The explanation of lab services (their statistical significance) is not quite convincing to me (based on my field observation and experience).

Response: See response to similar comment above. We have added some additional language to the discussion section to more clearly compare/contrast our study with other studies on peer educators for diabetes patients. As for our explanation of lab services, we have added another possible explanation to the discussion section, as below.

“It is possible that participants who utilize laboratory tests are more likely to have good glycemic control due to another shared characteristic, such as higher socioeconomic status given the additional cost.”

I am not quite clear of the "Greater than zero but less than one laboratory test per year". What does it mean?

Response: On average during the follow-up period, this group had less than one laboratory test per year, but greater than zero tests. For example, a participant with 3 laboratory tests during a 4 year period had 3 / 4 = 0.75 lab tests per year. This group had more tests than those who had exactly zero tests during the follow-up period.

---

## [Decision Letter · Decision Letter 1]

29 May 2020

PONE-D-20-00770R1

Utilization of diabetes management health care services and its association with glycemic control among patients participating in a peer educator-based program in Cambodia

PLOS ONE

Dear Dr. Rao,

Thank you for submitting your manuscript to PLOS ONE. After careful consideration, we feel that it has merit but does not fully meet PLOS ONE’s publication criteria as it currently stands. Therefore, we invite you to submit a revised version of the manuscript that addresses the points raised during the review process.

We look forward to receiving your revised manuscript.

Kind regards,

Siyan Yi, MD, MHSc, PhD

Academic Editor

PLOS ONE

**Comments to the Author**

1. If the authors have adequately addressed your comments raised in a previous round of review and you feel that this manuscript is now acceptable for publication, you may indicate that here to bypass the “Comments to the Author” section, enter your conflict of interest statement in the “Confidential to Editor” section, and submit your "Accept" recommendation.

Reviewer #1: All comments have been addressed

Reviewer #2: All comments have been addressed

Reviewer #3: All comments have been addressed

2. Is the manuscript technically sound, and do the data support the conclusions?

Reviewer #1: Yes

Reviewer #2: Yes

Reviewer #3: Yes

3. Has the statistical analysis been performed appropriately and rigorously? 

Reviewer #1: Yes

Reviewer #2: Yes

Reviewer #3: Yes

4. Have the authors made all data underlying the findings in their manuscript fully available?

Reviewer #1: Yes

Reviewer #2: Yes

Reviewer #3: Yes

5. Is the manuscript presented in an intelligible fashion and written in standard English?

Reviewer #1: Yes

Reviewer #2: Yes

Reviewer #3: Yes

6. Review Comments to the Author

Reviewer #1: I think the authors have appropriately addressed all my previous comments, which made the revised manuscript in line with academic standard and ready for publication. Findings from this study would help strengthening the current evidence to support beneficial impact of peer educators on diabetes care, particularly in deprived settings.

Reviewer #2: The authors have addressed the comments by making substantial changes to the title, objective, discussion and conclusion.

Reviewer #3: Thanks for carefully addressing my comments in the previous round, and I am satisfied with your responses. I understand that some limitations cannot be overcome. With this revised version, you have acknowledged those limitations, making the conclusion acceptable. However, I still have three minor comments.

1. On page 5 (line 80), please give a recent update of the coverage.

2. On page 6 (line 100), when you said a "small fee", to which standard do you compare (high income living standard?)? In Cambodian rural areas, this amount, to me, may not be small.

3. On page 13 (line 233), do you have any reference for point '4' saying medication adherence could not be accurately calculated in these instances?

7. PLOS authors have the option to publish the peer review history of their article (what does this mean?). If published, this will include your full peer review and any attached files.

Reviewer #1: Yes: Teerapon Dhippayom

Reviewer #2: No

Reviewer #3: No

---

## [Author Response · Author response to Decision Letter 1]

2 Jun 2020

Reviewer #1: I think the authors have appropriately addressed all my previous comments, which made the revised manuscript in line with academic standard and ready for publication. Findings from this study would help strengthening the current evidence to support beneficial impact of peer educators on diabetes care, particularly in deprived settings.

Response: none

Reviewer #2: The authors have addressed the comments by making substantial changes to the title, objective, discussion and conclusion.

Response: none

Reviewer #3: Thanks for carefully addressing my comments in the previous round, and I am satisfied with your responses. I understand that some limitations cannot be overcome. With this revised version, you have acknowledged those limitations, making the conclusion acceptable. However, I still have three minor comments.

1. On page 5 (line 80), please give a recent update of the coverage.

Response: We have added an update to the number of peer educators trained and participants served.

2. On page 6 (line 100), when you said a "small fee", to which standard do you compare (high income living standard?)? In Cambodian rural areas, this amount, to me, may not be small.

Response: We have added a sentence to demonstrate that MoPoTsyo’s fees represent a small percentage of monthly household income in Cambodia. If this does not suffice, we can remove the word “small.”

3. On page 13 (line 233), do you have any reference for point '4' saying medication adherence could not be accurately calculated in these instances?

Response: We used the specifications established by the National Quality Forum for calculating Proportion of Days Covered, which states that the treatment period must be at least 90 days to calculate this measure. We have added a reference for this.

---

## [Editor Report · Decision Letter 2]

9 Jun 2020

Utilization of diabetes management health care services and its association with glycemic control among patients participating in a peer educator-based program in Cambodia

PONE-D-20-00770R2

Dear Dr. Rao,

We’re pleased to inform you that your manuscript has been judged scientifically suitable for publication and will be formally accepted for publication once it meets all outstanding technical requirements.

Kind regards,

Siyan Yi, MD, MHSc, PhD

Academic Editor

PLOS ONE
---

## [Editor Report · Acceptance letter]

17 Jun 2020

PONE-D-20-00770R2 

Utilization of diabetes management health care services and its association with glycemic control among patients participating in a peer educator-based program in Cambodia 

Dear Dr. Rao:

I'm pleased to inform you that your manuscript has been deemed suitable for publication in PLOS ONE. Congratulations! Your manuscript is now with our production department. 

Kind regards, 

on behalf of

Dr. Siyan Yi 

Academic Editor

PLOS ONE